# Exogenous MnSO_4_ Improves Productivity of Degenerated *Volvariella volvacea* by Regulating Antioxidant Activity

**DOI:** 10.3390/jof10120825

**Published:** 2024-11-27

**Authors:** Qiaoli Wang, Wenpei Wang, Yonghui Wang, Jinmin Yun, Yubin Zhang, Fengyun Zhao

**Affiliations:** 1College of Food Science and Engineering, Gansu Agricultural University, Lanzhou 730070, China; 18409483023@163.com (Q.W.); 18152275316@163.com (Y.W.); yunjianmin@gsau.edu.cn (J.Y.); zhangyb@gsau.edu.cn (Y.Z.); 2Kangle County Special Agricultural Development Center, Linxia 731599, China; 3Lanzhou Bioproducts Research Institute, Lanzhou 730046, China; w18419167728@163.com

**Keywords:** *Volvariella volvacea*, MnSO_4_, strain rejuvenation, lignocellulase, antioxidant activity, productive characteristics

## Abstract

Manganese is one of the trace elements necessary for organisms to maintain normal biological activities and is also a cofactor for manganese superoxide dismutase (Mn-SOD) and manganese peroxidase (MnP). In order to find a simple and effective method to rejuvenate the degenerated *V. volvacea* strains, we explored the effect of the exogenous addition of MnSO_4_ on the antioxidant vigour and productivity of degenerated strains of *V. volvacea*. The results showed that the exogenous MnSO_4_ had no significant effect on the non-degenerated strain T0, but it could effectively increase the mycelial growth rate, mycelial biomass, and LBL decolouring ability of the degenerated strains T10 and T19, and reduce the production cycle and increased the biological efficiency of T10; it helped the severely degenerated T19 to regrow its fruiting body; and it also significantly increased the viability of the matrix-degrading enzymes such as EG, Lac, Xyl, etc. of T10 and T19. Meanwhile, exogenous MnSO_4_ significantly increased the activity of GPX, GR, CAT, SOD, and the content of GSH, polyphenols, minerals, and polysaccharides in T10 and T19 strains, which resulted in a significant decrease in the accumulation of ROS, such as O_2_^−^ and H_2_O_2_ in T10 and T19. The correlation analysis showed that there was a significant correlation between antioxidant activity and the production ability of *V. volvacea*. This study can provide theoretical reference and technical support for the rejuvenation research of degenerated strains of *V. volvacea* and other edible fungi.

## 1. Introduction

*Volvariella volvacea* (*V. volvacea*) is one of the important export and foreign-exchange-earning mushrooms in China, which is not only rich in edible value but also has good medicinal value. As a grass-rotting fungus, the ecological cycle is facilitated by *V. volvacea* secretion of substrate-degrading enzymes that promote the breakdown of agricultural waste, thereby acquiring the nutrients essential for its growth and development, and at the same time, it also produces a variety of nutrients that are beneficial to human health, such as polysaccharides, proteins, and mineral elements, etc. [1]. Like other edible fungi, strain degeneration is an important factor limiting the development of the *V. volvacea* industry. Strain degeneration is mainly characterised by the slow growth rate of mycelia, poor ability of mycelia to degrade the matrix, delayed formation of the fruiting body, inconspicuous mushrooming tides, declining yields, or even not growing into fruiting bodies, which has brought about a great impact on the development of the *V. volvacea* industry [2].

In production, asexual reproduction methods such as mycelial succession and tissue isolation succession are often used to maintain the viability of strains. However, long-term and multiple asexual succession can also lead to the degeneration of *V. volvacea* strains [3]; the low-temperature storage method is an effective measure to delay the degeneration of strains [4]. However, *V. volvacea* is a high-temperature type fungus, which is not resistant to low-temperature storage. The mycelia autolyse and die within 48 h at 4 °C, so its degeneration phenomenon is particularly serious [5]. Therefore, the development of a convenient and efficient strain rejuvenation technology is a key technical problem that needs to be solved urgently for the development of the *V. volvacea* industry.

Mineral elements are important nutrients in microbiological media and cofactors or activators of many enzymes in organisms, and 50–70% of enzymes require the involvement of mineral elements for activation in order to catalyse relevant biological and chemical reactions [6]. Mn is one of the trace elements essential for organisms to maintain normal life activities. In plant organisms, Mn has an important influence on their photosynthesis, maintenance of normal structure of organelles, and activation of enzyme activities [7]. Most fungi mitochondria contain superoxide dismutase molecules with manganese as a cofactor and have an activating effect on catalase, which scavenges superoxide anion radicals. Mn also has a role in the synthesis, catabolism, and respiration of substances in edible mushrooms, which in turn promotes the growth and development of the fungi [8]. Dhaliwal et al. showed that the use of both MnSO_4_ and MnCO_3_ significantly improved the growth, yield, and Mn uptake in wheat over the control [9]. Ma and Wang found that Mn treatment promoted the expression of the Glycyrrhiza uralensis Fisch SQS1 gene and the accumulation of glycyrrhiza uralensis [10]. Estrada and Royse investigated the effect of two mineral elements Mn and Cu on the growth of almond abalone mushrooms. It was found that the addition of Mn^2+^ could better promote mycelial growth and increase the production of substances [11].

In this paper, the degenerated strains of *V. volvacea* T10 and T19 and the original strain of *V. volvacea* V844 (T0) obtained in the preliminary stage of the group were used as test strains. MnSO_4_ was added exogenously in PDA and the culture matrix to study the rejuvenation effect of MnSO_4_ on the degenerated strains of *V. volvacea* and the rejuvenation mechanism.

## 2. Materials and Methods

### 2.1. Strains and Media

The original strain of *V. volvacea* (T0), a commercialized cultivated strain V844, was conserved in the College of Food Science and Engineering, Gansu Agricultural University, China.

*V. volvacea* subcultured strains (T10 and T19) were obtained by repeated tissue isolation [12]. Briefly, T0 was cultivated, and an egg-shaped stage fruiting body was obtained. The fruiting body was cut, and a small piece from the junction of the stipe and the cap was used to generate the 1st generation strain, labelled as T1. T1 was cultivated, and the obtained fruiting body was used to obtain the 2nd generation strain, labelled as T2. Likewise, T1–T19 strains were obtained using 19 consecutive cultivations. T10 and T19 were selected as experimental strains. All strains were stored in liquid paraffin at a constant temperature of 20 °C.

PDA medium: potato 20.0%, glucose 2.0%, KH_2_PO_4_ 0.1%, MgSO_4_ 0.1%, agar powder 2.0%. Seed medium (*w*/*w*): cottonseed husk 88%, bran 10%, gypsum 1%, lime 1%, water content 65%, pH 8–9. cultivation medium (*w*/*w*): waste cotton 95%, lime 5%, water content about 65%, pH 9–10.

Liquid medium supplemented with bromothymol blue and lactose (LBL): potato 20%, lactose 2.0%, NH_4_NO_3_ 0.2%, KH_2_PO_4_ 0.15%, MgSO_4_ 0.05%, bromothymol blue (BTB) (Shanghai Blue season technology development Co., Ltd., Shanghai, China) 0.006%.

MnSO_4_ (Shanghai Yuanye Biotechnology Co., Ltd., Shanghai, China) treatment group: the optimum concentration of MnSO_4_ was added to the PDA medium; the water used for seed medium preparation was replaced with an aqueous solution of the optimum concentration of MnSO_4_. For cultivation management, the water used for spraying was replaced with an aqueous solution of the optimum concentration of MnSO_4_.

### 2.2. Screening of Optimal Concentration of MgSO_4_

Based on the PDA medium, different concentrations of MnSO_4_ (0, 25, 50, 75, and 100 mg/L) were added. The T10 strains were uniformly activated, received on the PDA medium, placed in a constant temperature at 30 °C (Shanghai Heng Scientific Instrument Co., Ltd., Shanghai, China), and then photographed, recorded, and observed after 72 h to screen out the optimal addition concentration.

### 2.3. Determination of Physiological Characters of V. volvacea Mycelia

Colony morphology: Referring to the method of Lyu et al. for determination [13], T0, T10, and T19 strains were inoculated with one mycelial block on PDA plates using a 6 mm diameter punch and cultured at a constant temperature of 30 °C for 72 h, then observed and photographed for recording.

Mycelial growth rate (mm/h): With reference to the method of Chung et al. [14], the T0, T10, and T19 strains were uniformly cultured at a constant temperature of 30 °C for 72 h, and the colony diameters were marked with a marking pen on the Petri dishes by the crosshatch method, and the mycelial growth rate was calculated according to the following equation.
(1)$$Mycelial\;growth\;rate\;(mm/h)=\frac{\left(colony\;diameter-inoculated\;mycelial\;block\;diameter\right)mm}{2\times 72\;h}\times 100\%$$

Mycelial biomass (g): Determined by referring to the method of Deshmukh and Bhaskaran [15], the cellophane was cut into a circle slightly smaller than the diameter of the Petri dish (9 mm) and sterilised, and the sterile cellophane was covered on the surface of the solidified PDA plate. The strains were inoculated on sterile cellophane and incubated at 30 °C for 72 h. The mycelial was gently scraped off with a slide, dried at 60 °C until constant weight and weighed.

LBL decolourisation ability: Refer to the method of Chen et al. [3]. Five pieces of mycelial agar with 1 cm diameter were inoculated into 100 mL LBL, and five pieces of the same blank agar were taken as a blank group. The culture was incubated in the dark at 30 °C and 110 rpm in shaking flasks for 6 d. The colour changes were observed and photographed for recording. Take an appropriate amount of fermented culture medium and determine the pH value of the medium with an acidimeter. The appropriate amount of culture broth was aspirated and centrifuged at 8000 rpm for 1 min, and the OD value of the supernatant was measured at 615 nm for the calculation of the decolourisation rate (expressed as a percentage).

The calculation formula is as follows:(2)$$Decolorizing\;rate\;of\;LBL\;medium\left(\%\right)=\frac{ODck-ODsample}{ODck}\times 100\%$$

ODck: OD values of the control group.

### 2.4. Determination of Traits of V. volvacea Fruiting Bodies

Referring to the method of Liu et al. [16], T0, T10, and T19 strains were inoculated into cultivation frames (40 cm × 20 cm × 10 cm) containing 1.2 kg of cultivation medium, and the cultivation baskets were covered with plastic bags and placed into the cultivation box at 30 °C as a constant temperature (Shanghai—Heng Scientific Instrument Co., Ltd., Shanghai, China). After the mycelia grew all over the frame, the plastic bag was removed, enough water was sprayed, and the lighting and humidifier were turned on to raise the temperature of the box to 33 °C, and the relative humidity of the air was kept at 85–90%. Three replicates were set for each strain, and the following indexes were determined during the cultivation process.

Primordium formation time (d): Record the time required from the start of cultivation to the formation of the first primordium.

Production cycle (d): Record the time required from the start of cultivation to the time when the substrate grows into the harvesting stage (egg-shaped stage).

Average weight of a fruiting body (g): Five egg-shaped stage substrates were randomly picked from each cultivation frame, weighed separately, and averaged.

Biological efficiency: Calculated according to the following formula
(3)$$Biological\;efficiency\left(\%\right)=\frac{\left(fresh\;fruiting\;body\;yield\right)}{\left(quantity\;of\;dry\;substrate\;used\right)}\times 100\%$$

### 2.5. Determination of Enzyme Activity Related to Matrix Degradation

Referring to the method of Long et al. for the determination [17], the activated *V. volvacea* mycelia were cultivated in a shaker at 33 °C and 180 rpm for 8 d. The appropriate amount of fermentation broth was taken and centrifuged at 4 °C and 10,000 rpm for 10 min, and the supernatant was taken as the crude enzyme solution. Then the filter paper activity (FPA), endoglucanase (EG), exoglucanase (CBH), β-glucosidase activity (BGL), laccase (Lac), manganese peroxidase (MnP), hemicellulase (HMC) and xylanase (Xyl) were determined using a commercial kit (Beijing Soleberg Technology Co., Ltd., Beijing, China) according to the manufacturer’s instructions.

### 2.6. Determination of Antioxidant Content

Crude polysaccharides were determined with reference to the method of Zhang et al. [18]; flavonoids and polyphenols were determined with reference to the method of Nie et al. [19]; and minerals were determined using a NovAA400P flame atomic absorption spectrometer (Jena, Germany). Glutathione (GSH) and oxidized glutathione (GSSG) contents were determined with reference to the method of Huang et al. [20], and then tested by using a commercial kit (Beijing Solepol Science and Technology Co., Ltd., Beijing, China) and assayed according to the manufacturer’s instructions.

### 2.7. Determination of ROS Content and Antioxidant Enzyme Activity

Referring to the method of Chen et al. [21], mycelium was collected by inoculating T0, T10, and T19 strains and culturing them in a potato dextrose broth (PDB) medium for 3 d. The cells were then incubated with a commercial kit (Beijing Solapur Science and Technology Co., Ltd., Beijing, China). The activity levels of intracellular hydrogen peroxide (H_2_O_2_), superoxide anion (O_2_^−^), superoxide dismutase (SOD), catalase (CAT), glutathione peroxidase (GPX), and glutathione reductase (GR) were then detected using a commercial kit (Beijing Solepol Science and Technology Co. Ltd., Beijing, China) according to the manufacturer’s instructions.

### 2.8. Data Processing

The determination of all indicators was repeated three times for each strain to obtain average values. Data were statistically processed using Microsoft Excel 2019 (Microsoft, Redmond, WA, USA), data were analysed for significance using IBM SPSS Statistics 27 (SPSS, Chicago, IL, USA) and plotted using Origin 2021 (Electronic Arts Inc., Chicago, IL, USA).

## 3. Results

### 3.1. Screening of Optimum Concentration of MnSO_4_ Addition

T10 was used as the primary screening strain, and MnSO_4_ was added in PDA at five concentration gradients (0 mg/L, 25 mg/L, 50 mg/L, 75 mg/L, and 100 mg/L), and the physiological traits of mycelium were used as the screening indexes. The results are shown in Figure 1. Compared with the CK group, the colony diameter showed a tendency to increase and then decrease with the increase of the added concentration of MnSO_4_, and the optimal added concentration was 50 mg/L.

### 3.2. Effect of MnSO_4_ on the Physiological Traits of V. volvacea Mycelia

The optimal concentration of 50 mg/L MnSO_4_ was exogenously added to the PDA medium to observe the colony morphology of T0, T10, and T19, and the results are shown in Figure 2. With the increase in the number of successions, the colony diameter of *V. volvacea* in the CK group showed a trend of gradual reduction, the colony diameter of T0 was the largest, and the colony diameter of T10 was the smallest. After the exogenous addition of the optimum concentration of MnSO_4_, the colony diameters of strains T0, T10, and T19 were significantly larger than those of the CK group.

The mycelial growth rate and mycelial biomass of strains T0, T10, and T19 were significantly increased by the addition of MnSO_4_; compared to the CK group, the mycelial growth rate of T0, T10, and T19 increased by 4.4%, 22.3%, and 27.4%, respectively. Mycelial biomass increased by 5.3%, 27.3%, and 30.2%, respectively. It can be seen that MnSO_4_ has a better recovery effect on the mycelial traits of degraded strains T10 and T19 of *V. volvacea*.

### 3.3. Effect of MnSO_4_ on the Decolourising Ability of V. volvacea LBL

The LBL decolourisation method can reflect the viability of the strains and the degree of degeneration of the strains to a certain extent. The decolourisation ability of the LBL medium of T0, T10, and T19 was determined, and the results are shown in Figure 3. The decolourisation ability of the T0 strain was stronger and the LBL medium was orange-yellow. There was no obvious change in the colour of the medium of the MnSO_4_-treated group and the CK group. Strain T19 had the worst decolourisation ability, and the colour of the medium in the CK group was dark green, and the decolourisation ability was significantly enhanced by MnSO_4_ treatment, and the colour of the medium was light yellow.

The decolourisation rate and pH of the medium were determined, and the results showed that the decolourisation rates of T0, T10, and T19 strains were increased by 6.0%, 23.9%, and 43.2%, respectively, after MnSO_4_ treatment (Figure 3B); and the pH values of the medium were decreased by 2.7%, 5.8%, and 16.9%, respectively (Figure 3B). Correlation analysis showed that the OD value and pH of the LBL medium showed a significant positive correlation (R^2^ = 0.98484) (Figure 3C).

### 3.4. Effect of MnSO_4_ on Agronomic Traits of V. volvacea Fruiting Bodies

Cultivation tests were carried out on T0, T10, and T19, and the results are shown in Appendix A. T0 and T10 were able to produce fruiting bodies successfully, while T19 could only form primordium and could not grow further into fruiting bodies. After MnSO_4_ treatment, the diameter and number of fruiting bodies of the T0 and T10 strains increased to different degrees, and the T19 strain, which could not produce fruiting bodies, regrew fruiting bodies.

The time of primordium formation, the production cycle, the average weight of a fruiting body, and the biological efficiency were further determined, and the results are shown in Figure 4. After MnSO_4_ treatment, the differences in primordia formation time, production cycle, average weight of fruiting body, and biological efficiency of T0 strain were not significant. The time of primordium formation of T10 and T19 strains was reduced by 14.7% and 20.2% compared with that of the control group, respectively; the production cycle of the T10 strain was shortened by 7.9%, the average weight of the fruiting body was increased by 5.8%, and the biological efficiency was increased by 46.0%. The MnSO_4_ treatment group caused strain T19 to produce fruiting bodies again with a production cycle of 28.2 d and a biological efficiency of 12.7%.

### 3.5. Effect of MnSO_4_ on the Activity of Matrix-Degrading Enzymes in V. volvacea

The cellulose degradation-related enzyme activities of strains T0, T10, and T19 were determined and the results are shown in Figure 5A–D. The enzyme activities of FPA, EG, CBH, and BGL showed a decreasing trend with the increase of the number of strain succession.

After the exogenous addition of MnSO_4_, the enzyme activities of FPA and EG of strain T0 were significantly increased by 20.0% and 13.2% (*p* < 0.05) compared with the control, and the differences in the enzyme activities of CBH and BGL were not significant; the enzyme activities of FPA, EG, BGL, and CBH of T10 were increased by 26.1%, 12.4%, 10.4%, and 6.4%, respectively, and the differences were significant (*p* < 0.05); FPA, EG, BGL, and CBH enzyme activities increased by 36.7%, 21.7%, 22.0%, and 18.8%, respectively, in T19, with significant differences (*p* < 0.05).

The enzyme activities related to lignin degradation were determined for T0, T10, and T19, and the results are shown in Figure 5E,F. Compared with T0, the Lac and MnP enzyme activities of T10 and T19 showed a decreasing trend.

The difference was not significant for T0 after the exogenous addition of MnSO4. The Lac and MnP activities of T10 increased by 12.0% and 23.5%, respectively, and those of T19 increased by 29.6% and 43.8%, respectively, which were both significantly (*p* < 0.05) higher than those of the control.

The enzyme activities related to hemicellulose degradation were determined for T0, T10, and T19, and the results are shown in Figure 5G,H. Compared with T0, the changes in HMC and Xyl enzyme activities were not significant. The exogenous addition of MnSO_4_ increased the HMC and Xyl enzyme activities of T19 by 9.6% and 6.8%, respectively, which were both significantly (*p* < 0.05) higher than the control.

### 3.6. Effect of MnSO_4_ on Antioxidant Enzyme Activity in V. volvacea

The antioxidant enzyme activities of strains T0, T10, and T19 were determined and the results are shown in Figure 6. The vigour of SOD, CAT, GPX, and GR showed a decreasing trend with the increase in the number of succession of *V. volvacea* strains.

After the exogenous addition of MnSO4, the four antioxidant enzyme activities of T0, T10, and T19 showed different degrees of increase. The difference in SOD activity of T0 was not significant, and the activities of CAT, GPX, and GR were increased by 4.5%, 30.6%, and 17.6%, respectively. The activities of SOD, CAT, GPX, and GR of T10 strains were significantly increased by 19.3%, 9.5%, 17.0%, and 15.9%, respectively, and the T19 strain significantly increased by 35.2%, 117.6%, 61.5%, and 91.2%, respectively (*p <* 0.05).

### 3.7. Effect of MnSO_4_ on the Content of Antioxidant Substances in V. volvacea

The antioxidant contents of T0, T10, and T19 strains were determined and the results are shown in Figure 7. The contents of GSH, GSSG, polyphenols, flavonoids, and polysaccharides showed a decreasing trend with the increase in the number of successions. The contents of GSH, GSSG, crude polysaccharides, polyphenols, and flavonoids of T19 were 30.1%, 19.9%, 5.2%, 28.3%, and 17.8% lower than that of T10, respectively.

After exogenous addition of MnSO4, there was no significant change (*p >* 0.05) in the GSH and GSSG contents of T0, whereas the GSH and GSSG contents of T10 strain were increased by 10.9%, 26.9%, and 46.9%, 56.0% in T19 strain compared to the control group, respectively. The exogenous addition of MnSO_4_ had no significant effect on flavonoid content. The polysaccharide content of T0, T10, and T19 increased by 0.5%, 0.8%, and 2.6%, respectively, and the polyphenol content increased by 1.3%, 2.1%, and 1.8%, respectively, compared with the control.

### 3.8. Effect of MnSO_4_ on the Mineral Content of V. volvacea

The nine mineral contents of strains T0, T10, and T19 were determined and the results are shown in Table 1. The mineral contents of each strain were in the order of K > Mg > Ca > Na > Fe > Cu > Zn > Mn > Se. With the increase in the number of successions, the mineral enrichment effect of the successor strains gradually weakened, and the contents of each mineral showed a decreasing trend with the prolongation of the successional time. The total amount of minerals of strain T19 decreased by 11.2% compared with that of T0.The content of Na was the most reduced, and the content of Na of T19 was 48.2% less than that of T0. The content of Mg decreased the least, and the content of Na of T19 was 48.2% less than that of T0. The Na content of strain T19 decreased by 48.2% compared with that of strain T0, and the Mg content decreased by 4.6% compared with that of strain T0.

After the exogenous addition of MnSO4, the nine mineral contents of T0, T10, and T19 strains showed some degree of increase. The K, Mg, Ca, Na, Fe, Cu, Zn, Mn, and Se contents of T0 strains were increased by 5.2%, 7.5%, 0.4%, 3.8%, 42.0%, 11.9%, 12.8%, and 175.5% compared with that of the control group, respectively, The K, Mg, Ca, Na, Fe, Cu, Zn, Mn, and Se contents of strain T10 were increased by 4.8%, 2.7%, 0.1%, 13.7%, 19.4%, 14.9%, 11.9%, 190.4%, and 57.9%, respectively, compared with that of the control group, Se content increased by 5.7%, 7.8%, 0.6%, 9.2%, 20.5%, 9.3%, 33.9%, 222.3%, and 93.3%, respectively, compared with the control.

### 3.9. Effect of MnSO_4_ on O_2_^−^ and H_2_O_2_ Content of V. volvacea

The O_2_^−^ and H_2_O_2_ contents of T0, T10, and T19 strains were determined, and the results are shown in Figure 8. The O_2_^−^ and H_2_O_2_ contents of the T0 strain were lower. The O_2_^−^ and H_2_O_2_ contents of T10 and T19 strains were significantly higher than those of T0.

After exogenous addition of MnSO_4_, the O_2_^−^ and H_2_O_2_ contents were significantly reduced (*p* < 0.05) compared with the CK group, and the O_2_^−^ content of T0, T10, and T19 were reduced by 11.4, 14.9%, and 19.4%, respectively, and the H_2_O_2_ content of T0, T10, and T19 were reduced by 5.4%, 20.1%, and 26.1%, respectively.

### 3.10. Correlation Between Main Characteristics of V. volvacea Correlation Analysis

Correlation analysis was carried out on various detection indicators of *V. volvacea*, and the results are shown in Figure 9. ROS (O_2_^−^, H_2_O_2_) was negatively correlated with antioxidant enzymes (SOD, CAT, GPX, and GR) and antioxidant substances (GSH, GSSG, flavonoids, polyphenols, polysaccharides, and minerals). The growth rate and biological efficiency of mycelia were positively correlated with antioxidant enzymes (SOD, CAT, GPX, and GR) and matrix-degrading enzymes (MnP, Lac, BGL, and CBH), and negatively correlated with ROS (O_2_^−^ and H_2_O_2_) content. The formation time and production cycle of primordia were positively correlated with ROS (O_2_^−^ and H_2_O_2_) content but negatively correlated with antioxidant enzymes (SOD, CAT, GPX, and GR), matrix-degrading enzymes (MnP, Lac, BGL, CBH), and antioxidant substances (GSH, GSSG, flavonoids, polyphenols, polysaccharides, and minerals).

### 3.11. Principal Component Analysis (PCA)

Principal component analysis mainly uses the idea of dimensionality reduction to transform multiple indicators into a few composite indicators to reflect the information in the original indicators, and when the cumulative contribution rate is higher than 60%, it indicates that the model is reliable. The higher the contribution rate of principal components, the better it can reflect the sample information. The results of the PCA on the comprehensive indicators such as mycelial characteristics, fruiting body traits, and nutrients of *V. volvacea* strains are shown in Figure 10. The variance contribution rate of the first principal component was 84.61%, and that of the second principal component was 8.89%, and the cumulative contribution rate reached 93.5%. It indicates that the extracted principal components are representative of the study of *V. volvacea* strains. Among them, 15 comprehensive indexes such as mycelial growth rate, mycelial biomass, biological efficiency, CAT, SOD, GPX, GR, MnP, CBH, BGL, EG, GSH, and crude polysaccharides, K and Zn, were mainly reflected in the first principal component, and 5 indexes such as Mn, Se, Cu, flavonoids, and polyphenols were mainly reflected in the second principal component; The distribution of ROS (O_2_^−^ and H_2_O_2_) content and other indexes were scattered, while the distribution of other indexes was more concentrated (Figure 10A). Figure 10B represents the scores of the samples of *V. volvacea*, and the six samples were divided into two parts, which were the control group of the succeeding strains of *V. volvacea* and the treatment group of externally added MnSO_4_. T10 MnSO_4_ > T10 CK, T19 MnSO_4_ > T19 CK. The results showed that the exogenous addition of MnSO_4_ was able to rejuvenate the degenerated strains of *V. volvacea*.

## 4. Discussion

The quality of strains directly affects the yield and quality of edible fungi, and frequent strain degeneration is a long-standing problem in edible fungi production [22]. There are many ways to rejuvenate degenerated strains, and changing the medium formulation is a simple and effective rejuvenation method [23]. Mineral elements are important nutrients in microbial media and play an extremely important role in the growth and development of edible fungi [24]. Włodarczyk et al. stated that the addition of zinc and magnesium salts to the cultivation medium increased the biomass of *Pleurotus spp.* mycelia [25]. The results of Ren et al. showed that the addition of Na_2_SeO_3_ enhanced the CAT, SOD, and GSH-PX activities of *Cordyceps militaris*, delayed the senescence rate, and increased the growth rate and the biomass of the mycelia [26]. Poursaeid et al. found that the addition of Zn to flat mushroom cultivation contributed to the increase of mycelial biomass and fruiting body production of *Pleurotus florida* [27]. Zhang et al. found that the addition of selenium had a significant effect on the growth of *Ganoderma lucidum* mycelia, morphological characteristics, yield, and activity substances, and the growth of Ganoderma lucidum mycelia with selenium added to the substrate was faster, which suggests that selenium promotes the growth of Ganoderma lucidum in the mycelial stage [28]. The results of this study showed that the exogenous addition of MnSO_4_ restored the production traits of *V. volvacea* to a certain extent, and the mycelial growth rate and biomass of degenerated strains of *V. volvacea* increased significantly (Figure 2), which led to a significant shortening of the production cycle and a significant increase in the biological efficiency of degenerated strain T10 of *V. volvacea*, and caused T19, which could not have produced fruiting bodies, to produce fruiting bodies again (Appendix A).

As one of the most important mineral elements in living organisms, manganese is involved in the synthesis and metabolism of many substances. Especially in the synthesis and action of enzymes. Manganese is an important component of superoxide dismutase and manganese peroxidase, and the synthesis of these enzymes is related to the scavenging of reactive oxygen species and the acquisition of nutrients in organisms; therefore, manganese is essential for organisms to cope with external stresses and to ensure their growth and development. In addition, manganese acts as a cofactor in enzymatic reactions and plays an irreplaceable role in the activation of more than 35 enzymes, including oxidoreductases, hydrolases, and convertases [29]. Schmidt and Husted showed that the crucial part of most enzymes in plant cells consists of manganese, and it is also an activator of many enzymes, and the appropriate amount of Mn can increase the activity of plant antioxidant enzymes, which ultimately affects the metabolism of the plant [30]. Weil et al. added manganese mineral powder to compost and significantly increased the yield of *Agaricus bisporus* mushrooms [31]. In the present study, the exogenous addition of MnSO_4_ resulted in enhanced mycelial vigour and increased fruiting body production in degenerated strains of *V. volvacea*.

As a typical grass-rotting fungus, *V. volvacea* has a strong matrix degradation ability. Its cultivation matrix is usually waste cotton, bran, cottonseed hulls, etc., which are composed of cellulose, hemicellulose, and lignin [32]. *V. volvacea* secrete relevant enzymes to degrade these difficult-to-break-down macromolecules into small molecules that they can easily absorb for their growth, development, and reproduction. The enzymes that degrade cellulose mainly include EG, CBH, and BGL. The breakdown of cellulose starts with the cleavage of long-chain cellulose into shorter oligosaccharides by EG, then further degradation to fibrous disaccharides by CBH, and finally to D-glucose by BGL [33]. The main enzymes for the degradation of hemicellulose are HMC and Xyl. The main component of hemicellulose is xylan, and Xyl hydrolyse long-chain xylan into short-chain oligosaccharides, which are further degraded into small molecules such as xylose [34]. Lignin degradation mainly includes Lac and MnP, which provide a carbon source for the growth of the fungus by breaking down the complex macromolecular lignin into small molecular compounds such as phenolic monomers. The level of enzyme activity of matrix-degrading enzymes measures the metabolic capacity of the mycelia, thus reflecting the ability of the strain to degrade the matrix and affecting the production cycle of edible fungi [35]. Zhao et al. found a significant decrease in ligninase, hemicellulase, and cellulase activity in degenerated *V. volvacea* strains [32]. Sun et al. showed that the degenerated strains of *Cordyceps sinensis* had reduced cellulase and amylase activities compared to normal strains [36]. The present study had a similar conclusion that the enzyme activities related to matrix degradation were also significantly reduced in *V. volvacea* strains after several successions. Cai et al. found that shiitake mushrooms are able to secrete lignin-degrading enzymes, hemicellulases, and cellulases that degrade lignocellulose into simple carbohydrates to support the growth of their own mycelia [37]. Fonseca et al. found that the exogenous addition of four metal compounds (CuSO_4_, AgNO_3_, MnSO_4,_ and CaCl_2_) was able to increase the ability of Lac activity of ligninolytic *Phlebia brevispora* BAFC 633 [38]. In this study, cottonseed husk was used as a matrix for the determination of related enzyme activities. The results showed that the exogenous addition of MnSO_4_ increased the activity of eight matrix degradation-related enzymes of degenerated strains T10 and T19 of *V. volvacea* to some extent (Figure 5). This may be due to the fact that Mn^2+^ is an essential factor for some enzymes, and the addition of Mn^2+^ can increase the activity of related degradation enzymes, which is conducive to the degradation of cottonseed hulls in the culture medium of edible fungi and provide nutrients for the growth of the mycelia of *V. volvacea*. Its specific mechanism needs to be further studied.

BTB in LBL medium is a textile dye derivative that turns yellow in an acidic environment and can be used as a pH indicator. In the process of mycelial culture, the strain metabolism produces organic acids, amino acids, CO_2,_ and other active substances, which are dissolved in the medium to produce free hydrogen ions, thus lowering the medium pH; when the strain degenerates its fermentation and metabolism ability is reduced, which in turn leads to the increase of medium pH and the decrease of decolouring ability [39,40]. Therefore, the degeneration degree of the strain can be identified by the decolourisation ability of the LBL medium. Magae et al. used the LBL decolourisation method to identify the degeneration of *Flammulina velutipes* and showed that the ability to decolourise the LBL medium of degenerated strains was significantly lower than that of normal strains. The present study had similar findings [41]. Meanwhile, Hoag et al. found that the degradation rate of BTB decreased with the increase of oxygen radicals [42]. In this study, it was found that the content of antioxidant substances such as antioxidant enzymes, polyphenols, flavonoids, and mineral elements was increased and the content of oxygen radicals such as O_2_^−^ and H_2_O_2_ were decreased in the degenerated strains of *V. volvacea* after the treatment with MnSO_4_, which may also be the reason for the significant decrease in the pH of the medium and the significant increase in the decolourisation ability of LBL in degenerated strains of *V. volvacea* with the addition of MnSO_4_ exogenously (Figure 3).

Reactive oxygen species (ROS) is a general term for oxygen-containing substances with active chemical properties and strong oxidation capacity. Aerobic organisms generate ROS during life activities from intracellular mitochondria and electron transport systems associated with the plasma membrane. Under physiological homeostatic conditions, ROS are detoxified through various antioxidant defence mechanisms [43]; however, excessive ROS generation can lead to oxidative damage, which can result in cellular senescence. Several studies have found that the degradation of edible mushroom strains is associated with excessive accumulation of ROS. Liu et al. found that in Ascomycetes, high ROS resulted in the gradual aging of the mycelia of *Cordyceps militaris* mycelia [44]. Xiong et al. found that the accumulation of intracellular ROS led to the degeneration of *Chrysophyllum chrysosporium* [45]. In this study, it was found that the content of reactive oxygen species such as O_2_^−^ and H_2_O_2_ were significantly increased in the strain with the increase of the degeneration degree of *V. volvacea* strains, and the exogenous addition of MnSO_4_ resulted in different degrees of reduction in the content of O_2_^−^ and H_2_O_2_ compared with the control group.

To resist oxidative damage, organisms have evolved a complex antioxidant defence system [46]. One is the antioxidant enzymes, involving antioxidant enzymes such as SOD, CAT, and GPX. The antioxidant enzyme system is effective in scavenging ROS. SOD is mainly responsible for scavenging active O_2_^−^ to H_2_O_2_ and CAT further decomposes H_2_O_2_ to O_2_ and H_2_O. H_2_O_2_ also oxidises GSH to produce GSSG in the presence of GPX, and GR is responsible for catalysing the reduction of GSSG to produce GSH [47], secondly, antioxidant substances (GSSG, GSH, flavonoids polyphenols, etc.) [48]. Yang et al. showed that the exogenous addition of salicylic acid could increase the activity of antioxidant enzymes and the content of antioxidants in jujube fruits, reduce the accumulation of H_2_O_2_ and O_2_^−^, and effectively reduce the respiratory intensity of the fruits, which could better maintain the hardness and colour of the fruits [49]. Taran et al. found that Zn stimulates the antioxidant defence system by reducing O_2_ or eliminating ROS to interfere with oxidative stress, thereby reducing H_2_O_2_ [50]. In the present study, the exogenous addition of MnSO_4_ increased SOD, CAT, GPX, GR activities and GSH, GSSG, flavonoids and polysaccharides contents of degenerated strains T10 and T19 to different extents, which was consistent with the changes in production trait indexes.

In this study, we preliminarily hypothesized that the possible reasons for MnSO_4_ restoring the production traits of degenerated strains of *V. volvacea* were that Mn^2+^ acted as a cofactor or activator of many enzymes in the organism, increased the activity of antioxidant enzymes and the content of antioxidant substances, and lowered the accumulation of ROS in the degenerated strains of *V. volvacea*, which in turn improved the scavenging ability of ROS. Moreover, Mn^2+^ could catalyse the activity of matrix-degrading enzymes in the degenerated strains of *V. volvacea*, enhance the decomposition and utilization of macromolecules in *V. volvacea*, provide glucose and other nutrients for the growth of *V. volvacea*, increase the mycelial growth rate and biomass of *V. volvacea*, as well as shorten the production cycle, restore the mushrooming ability of *V. volvacea*, improve the biological efficiency, and revive the production traits of the degenerated strains of *V. volvacea* (Figure 11).

## 5. Conclusions

This study explored the rejuvenation of *V. volvacea* degenerated strains by the exogenous addition of MnSO_4_. The results revealed that exogenous MnSO_4_ did not have a significant effect on the non-degenerated strain T0, but it could effectively improve the activity of the mycelia of degenerated strains T10 and T19 and significantly increased the activity of matrix-degrading enzymes of T10 and T19, which led to the shortening of the production cycle and the improvement of the biological efficiency of T10; it enabled T19 to regrow its fruiting bodies. Meanwhile, exogenous MnSO_4_ significantly increased the activity of antioxidant enzymes and the content of antioxidant substances in T10 and T19 strains, which led to a significant decrease in the accumulation of ROS in T10 and T19. Correlation analysis showed that there was a significant correlation between antioxidant activity and the production capacity of *V. volvacea*. It was found that exogenous MnSO_4_ sulphate played a crucial role in the growth, development, and yield of degenerated strains of *V. volvacea*.

## Figures and Tables

**Figure 1 jof-10-00825-f001:**
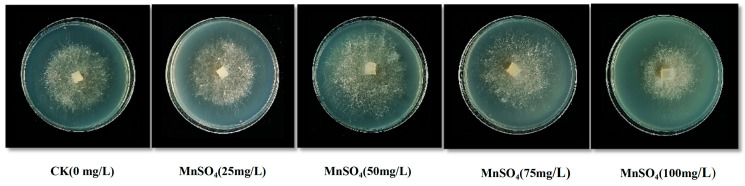
Effect of different concentrations of MnSO_4_ on the colony diameter of strain T10.

**Figure 2 jof-10-00825-f002:**
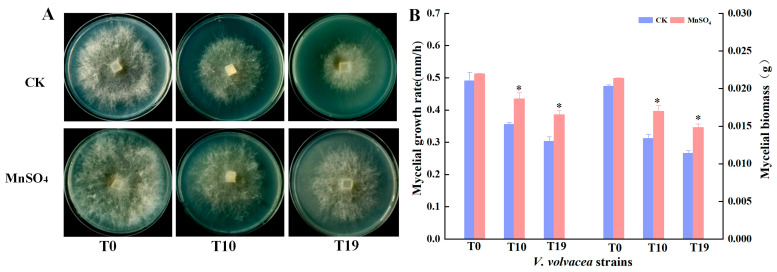
Effect of MnSO_4_ on the morphology (**A**), mycelial growth rate and biomass (**B**) of *V. volvacea* colonies. * represents a significant difference within the same group (*p* < 0.05).

**Figure 3 jof-10-00825-f003:**
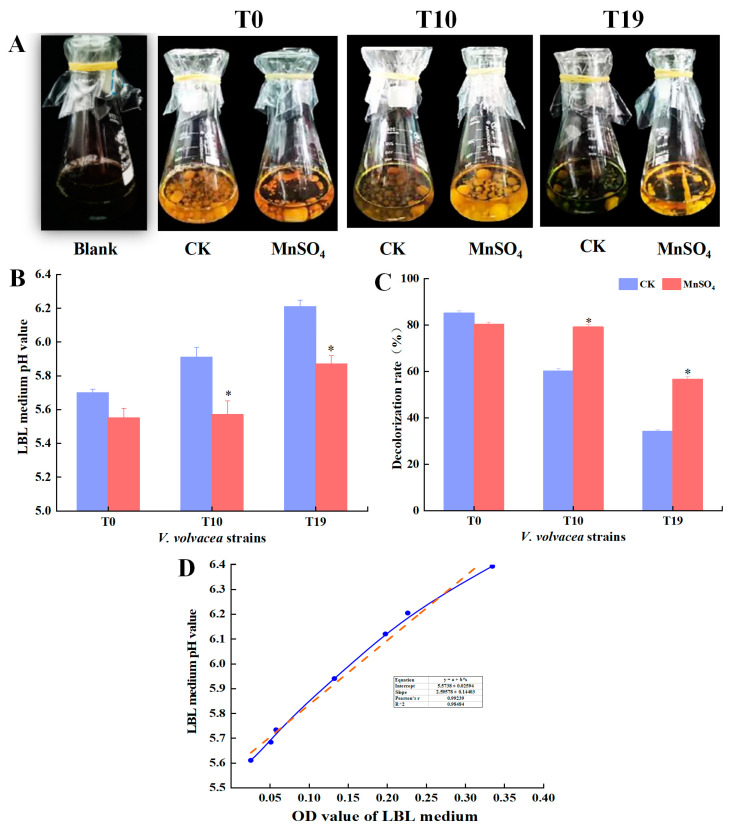
Effect of MnSO_4_ on decolourization ability of *V. volvacea* LBL. (**A**): Medium colour, (**B**): Medium pH, (**C**): Decolourization rate, (**D**): Correlation between pH and OD value, orange line: Linear fit of B “LBL medium pH”, blue line: LBL medium pH. * represents a significant difference within the same group (*p* < 0.05).

**Figure 4 jof-10-00825-f004:**
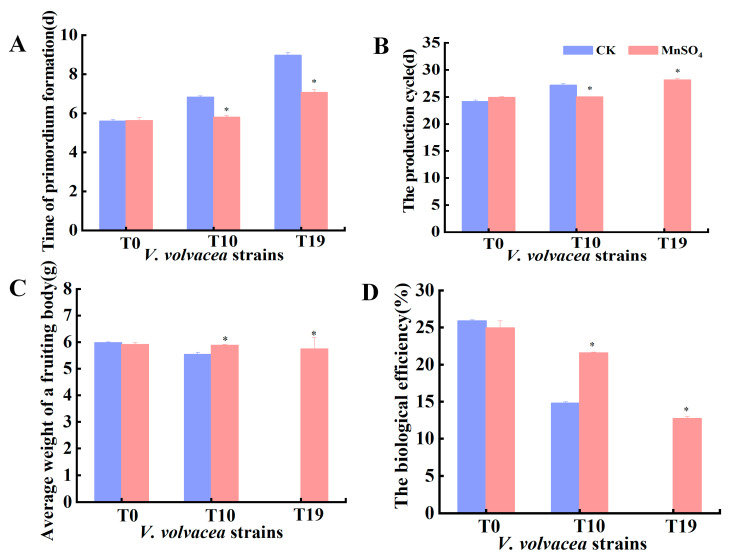
Effect of MnSO_4_ on production traits of *V. volvacea*. (**A**): Time of primordia formation, (**B**): The production cycle, (**C**): Average weight of a fruiting body, (**D**): The biological efficiency. * represents a significant difference within the same group (*p < 0.05*).

**Figure 5 jof-10-00825-f005:**
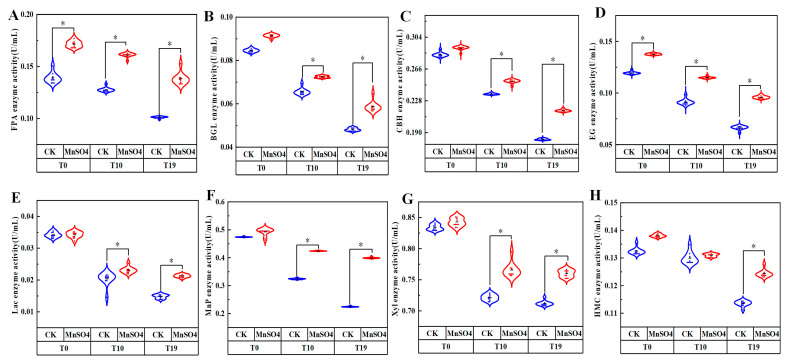
Effect of MnSO_4_ on matrix-degrading enzyme activity. (**A**): FPA, (**B**): BGL, (**C**): CBH, (**D**): EG, (**E**): Lac, (**F**): MnP, (**G**): Xyl, (**H**): HMA.CK: control group, MnSO_4_: MnSO_4_ treatment group, * represents a significant difference within the same group (*p* < 0.05).

**Figure 6 jof-10-00825-f006:**
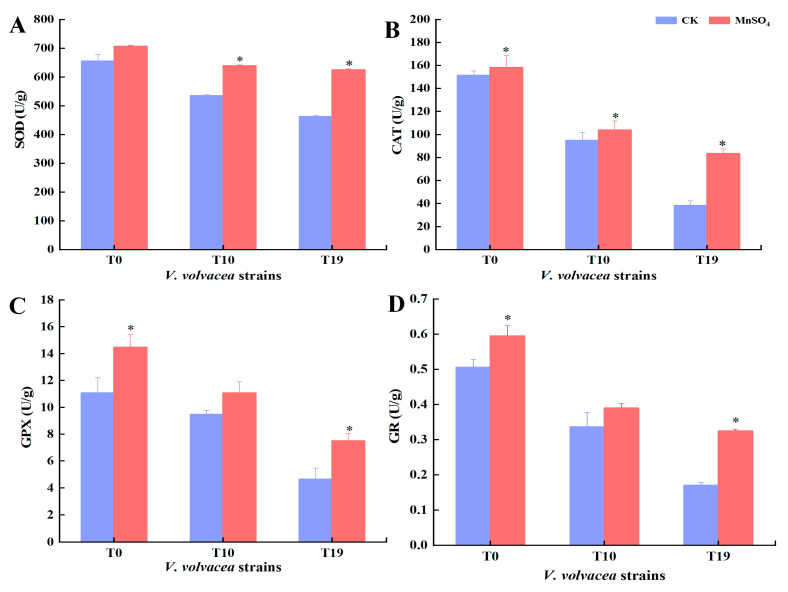
Effect of MnSO_4_ on antioxidant enzyme activity of *V. volvacea* mycelia.(**A**): SOD, (**B**): CAT, (**C**): GPX, (**D**): GR. * represents a significant difference within the same group (*p* < 0.05).

**Figure 7 jof-10-00825-f007:**
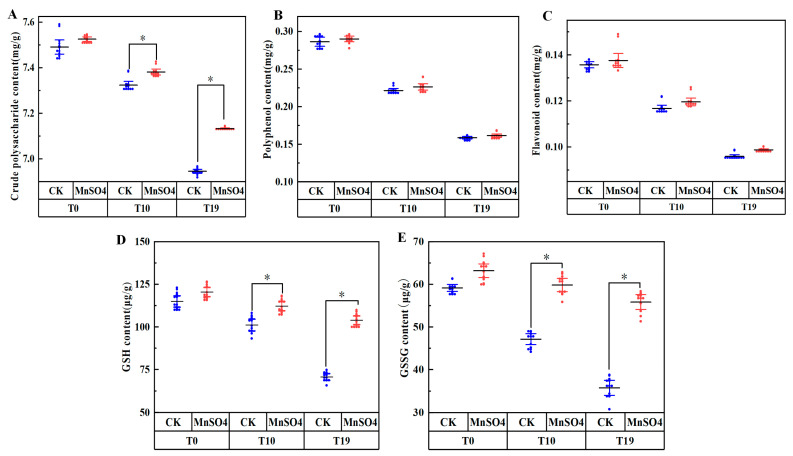
Effect of MnSO_4_ on crude polysaccharide (**A**), polyphenol (**B**) and flavonoid contents (**C**), GSH (**D**), GSSG (**E**) of *V. volvacea* mycelia. CK: control group, MnSO_4_: MnSO_4_ treatment group, * represents a significant difference within the same group (*p* < 0.05).

**Figure 8 jof-10-00825-f008:**
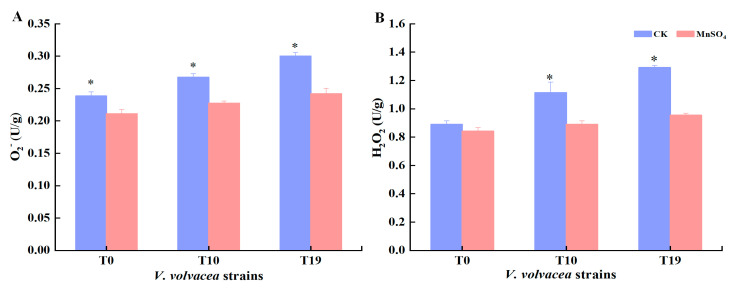
Effect of MnSO_4_ on O_2_^−^ (**A**) and H_2_O_2_ (**B**) contents of *V. volvacea* mycelia. * represents a significant difference within the same group (*p* < 0.05).

**Figure 9 jof-10-00825-f009:**
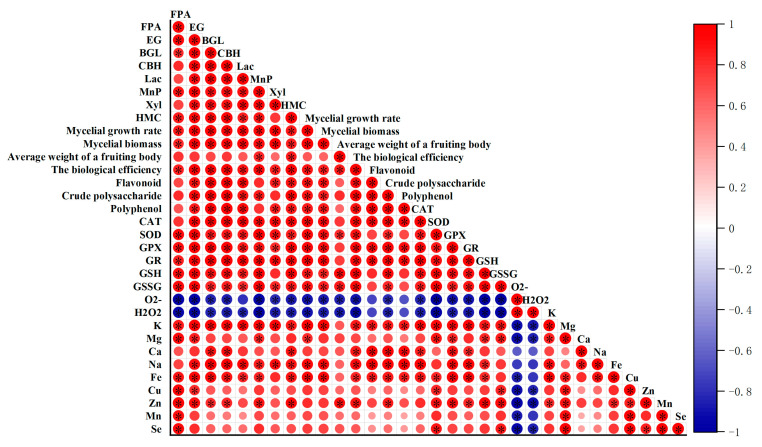
Correlation analysis between indicators of *V. volvacea* strains. * represents a significant difference within the same group (*p* < 0.05).

**Figure 10 jof-10-00825-f010:**
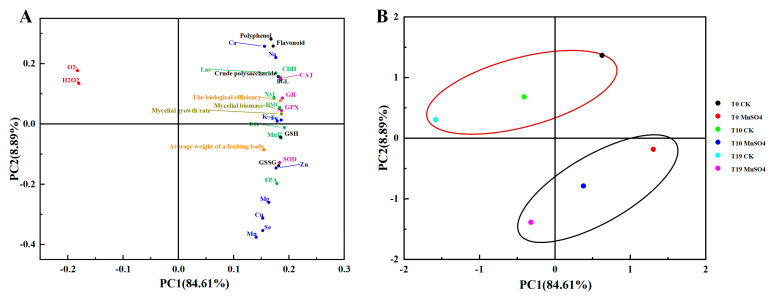
Principal component analysis of *V. volvacea* samples. (**A**): Loading plot, (**B**): Score plot.

**Figure 11 jof-10-00825-f011:**
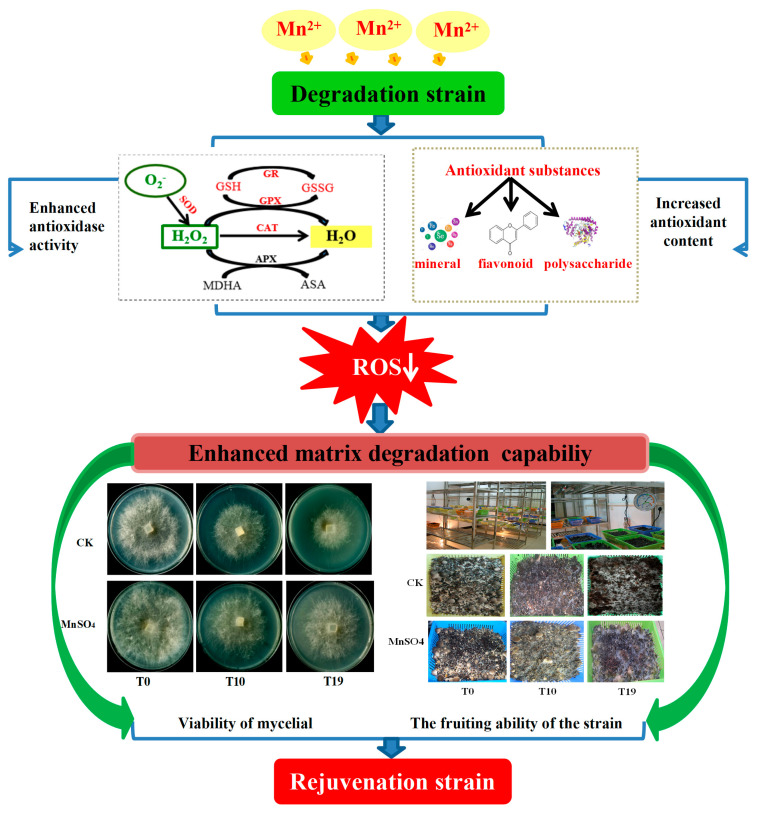
Pathway diagram for the revitalization of degenerated strains by exogenous addition of MnSO_4_.

**Table 1 jof-10-00825-t001:** Effect of MnSO_4_ on the mineral content of *V. volvacea*.

Mineral Contents	T0		T10		T19	
	CK	MnSO_4_	CK	MnSO_4_	CK	MnSO_4_
K/(mg/kg)	1117.57 ± 0.50	1176.13 ± 0.08	1035.44 ± 0.11	1084.63 ± 0.01	999.38 ± 0.02	1057.06 ± 0.01
Mg/(mg/kg)	675.86 ± 0.02	726.63 ± 0.02	673.56 ± 0.01	691.88 ± 0.01	645.06 ± 0.01	695.17 ± 0.29
Ca/(mg/kg)	161.15 ± 0.02	161.72 ± 0.01	157.71 ± 0.06	157.87 ± 0.01	139.77 ± 0.05	140.55 ± 0.04
Na/(mg/kg)	88.26 ± 0.02	91.58 ± 0.02	62.13 ± 0.02	70.67 ± 0.01	45.69 ± 0.01	49.89 ± 0.01
Fe/(mg/kg)	52.42 ± 0.01	74.44 ± 0.01	45.79 ± 0.01	54.69 ± 0.01	33.10 ± 0.00	39.89 ± 0.02
Cu/(mg/kg)	15.35 ± 0.02	17.18 ± 0.01	14.75 ± 0.04	16.95 ± 0.02	14.49 ± 0.01	15.84 ± 0.02
Zn/(mg/kg)	10.75 ± 0.01	12.13 ± 0.06	10.54 ± 0.02	11.79 ± 0.02	7.89 ± 0.01	10.57 ± 0.01
Mn/(mg/kg)	2.57 ± 0.02	7.08 ± 0.01	2.09 ± 0.01	6.07 ± 0.02	1.48 ± 0.01	4.77 ± 0.01
Se/(mg/kg)	0.22 ± 0.01	0.35 ± 0.01	0.19 ± 0.01	0.30 ± 0.01	0.15 ± 0.01	0.29 ± 0.02
Total (mg/kg)	2124.16 ± 0.61	2267.24 ± 0.21	2002.19 ± 0.29	2094.85 ± 0.10	1887.01 ± 0.01	2014.02 ± 0.42

## Data Availability

The raw data supporting the conclusions of this article will be made available by the authors on request.

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
