# Peer review of "Exogenous MnSO4 Improves Productivity of Degenerated Volvariella volvacea by Regulating Antioxidant Activity"

_jof, 2024, doi:10.3390/jof10120825_

Round 1
Reviewer 1 Report
Major comments
1. The aim of the work is not stated in the text.
2. When studying the growth rate, the authors measured the colony diameter and used this value to calculate the mycelial growth rate. This is incorrect, since the colony grows from the center in all directions and therefore, using the diameter, the authors
take the growth into account twice. The colony radius should be used.
3. The term "tissue" is used several times in relation to the fungus (L 42, 79-81). As is known, fungi do not have tissues.
4. In the "Materials and Methods" section, it is necessary to write what composition of the solid-phase substrate was used (L 130).
Minor comments
1. The sentence in L 62 is missing a verb.
2. In L 67, the word "substrates" is used. Apparently, the authors meant "substances".
3. Figure 3B is better presented as two figures.
4. There seems to be a typo in L 223 ("Figure S1").
5. In L 224 and 230, the word "protoplast" is mistakenly used instead of the word "primordium".
6. In L 272, the sentence begins with a lowercase letter.
7. In L 335, the word "primordials" is used instead of "primordia".
8. In L 353, it is written "Crude polysaccharide" instead of "Crude polysaccharides"

Author Response
We are grateful to you for comments and suggestions. Below we detail our responses to your comments and list the alterations we have made to the manuscript. Thank you very much for your valuable comments and constructive suggestions, which are vital to improve the quality of our manuscript. The revised parts are shown in red in our revised manuscript.
Comment 1: The aim of the work is not stated in the text.
Response 1: Thank you for your valuable advice, the expert opinion has been adopted.We have supplemented the relevant content in the manuscript.(L13-14)
Comment 2: When studying the growth rate, the authors measured the colony diameter and used this value to calculate the mycelial growth rate. This is incorrect, since the colony grows from the center in all directions and therefore, using the diameter, the authors take the growth into account twice. The colony radius should be used.
Response 2:Thank you for your valuable advice.the expert opinion has been adopted.We have corrected the formula and modified the figure.(L211-212)
The specific procedure for the determination of mycelial growth rate was as follows: after uniform activation of the strains, 1 cm diameter mycelial blocks were inoculated in the center of the PDA plates with a punch, and incubated at 30 ℃ for 72 h. The diameters of the V. volvacea colonies were marked on the backs of the plates with a marking pen by the crosshatch method. Because the measurement process was not photographed, we drew a schematic diagram on the front of the plate, and the starting and ending points of the line were shown in Figure. 1. In order to avoid the measurement error caused by the rapid growth of mycelial on one side, we took the average length of two lines AB and CD by cross-hatching to calculate the mycelial growth rate.
Figure.1 Schematic diagram of delineation in mycelial growth rate determination
Comment 3:The term "tissue" is used several times in relation to the fungus (L 42, 79-81). As is known, fungi do not have tissues.
Response 3: Thank you for your comment.
In practical production, tissue isolation technique is a common method for purification and rejuvenation of edible mushroom strains(Sun et al., 2017; Jin et al., 2023).V. volvacea subcultured strains (T10 and T19) were obtained through the successive tissue isolation subculturing method. Briefly, T0 was cultivated, and an egg-shaped stage fruiting body was obtained. The fruiting body was cut, and a small piece from the junction of the stipe and the cap was used to generate the 1st generation strain, labeled as T1. T1 was cultivated, and the obtained fruiting body was used to obtain the 2nd generation strain, labeled as T2. Likewise, T1-T19 strains were obtained using 19 consecutive cultivations(Figure 2). T10 and T19 were selected as experimental strains.Therefore, the "tissue" that appears in the manuscript is the egg-shaped stage fruiting body.
Figure 2. The tissue isolation process for succession strains is shown in corresponding images. (A) Original strains (T0), (B) Activation of strains, (C) Seed cultivation, (D) The original base, (E) Egg-shaped period, (F), Tissue separation of the fruiting body, (G) Tissue culture, (H) Strains preservation (T1-T19).
Sun, SJ; Deng, CH; Zhang, LY; Hu, KI. Molecular analysis and biochemical characteristics of degenerated strains of Cordyceps militaris.[J].Archives of Microbiology.2017,Vol.199(No.6):939-944.
Jin, L; Li, B; Tian, F;Shang, XX; Jia, MM.Comparative Experiment on Efficient Tissue Isolation of Dried Phellinus igniarius.[J].Northern Horticulture.2023,(01):113-118.
Comment 4:In the "Materials and Methods" section, it is necessary to write what composition of the solid-phase substrate was used (L 130).
Response 4: Thank you for your valuable advice, the expert opinion has been adopted.The composition of the solid-phase substrate in Materials and Methods section 2.1.(L93-95)
Comment 5:The sentence in L 62 is missing a verb.
Response 5:Thank you for your comment.
This has been corrected in the revised manuscript(L 67).
Comment 6: In L 67, the word "substrates" is used. Apparently, the authors meant "substances".
Response 6:Thank you for your comment.
This has been corrected in the revised manuscript(L 71).
Comment 7: Figure 3B is better presented as two figures.
Response 7:Thank you for your valuable advice.the expert opinion has been adopted.(L 230-231)
Comment 8: There seems to be a typo in L 223 ("Figure S1").
Response 8:Thank you for your comment.
In the manuscript Figure S1 is our supplementary figure.(L 236).
Comment 9: In L 224 and 230, the word "protoplast" is mistakenly used instead of the word "primordium".
Response 9:Thank you for your comment.
This has been corrected in the revised manuscript(L 237 and 243 ).
Comment 10: In L 272, the sentence begins with a lowercase letter.
Response 10:Thank you for your comment.
This has been corrected in the revised manuscript(L 285 ).
Comment 11: In L 335, the word "primordials" is used instead of "primordia".
Response 11:Thank you for your comment.
This has been corrected in the revised manuscript(L 348).
Comment 12: In L 353, it is written "Crude polysaccharide" instead of "Crude polysaccharides".
Response 12:Thank you for your comment.
This has been corrected in the revised manuscript(L 367).
Thank you so much!

Reviewer 2 Report
The manuscript entitled:"Exogenous MnSO4 improves productivity of degenerated Volvariella volvacea by regulating antioxidant activity" deals with the possibilities of rejuvenating the degenerate strains of the edible mushroom Volvariella volvacea by exogenously adding optimal concentrations of MnSO4 to the growth medium. The authors studied the effect of this compound on the physiological traits, the decolourising ability, agronomic characteristics, the activity of matrix-degrading enzymes, antioxidant enzyme activity, the content of antioxidant substances, O2- and H2O2 content, and the mineral content of V.volvacea (mycelium or fruiting bodies).
The results obtained from this study are very promising in terms of supporting research on the regrowth of damaged/weakened/degenerate strains of V. volvacea and other edible mushrooms under industrial cultivation conditions. It happens that unforeseen circumstances appear that result in the loss of the desired characteristics of the production strains, which ultimately causes an inadequate or absent mushroom yield, and thus great economic losses. Considering the increasing needs of the growing human population for food sources, especially healthy foods such as mushrooms, such research is very important as it could contribute to more efficient production.
All my sugestions are attached.

Author Response
Author’s Responses to Reviewers’ Comments
We are grateful to you for comments and suggestions. Below we detail our responses to your comments and list the alterations we have made to the manuscript. Thank you very much for your valuable comments and constructive suggestions, which are vital to improve the quality of our manuscript. The revised parts are shown in red in our revised manuscript.
Comment 1:In L 13, replace the word “paper” with “research”.
Response 1: Thank you for your valuable advice, the expert opinion has been adopted.(L14)
Comment 2:"...an excellent biological reactor.During the growth process,V.volvacea use the matrix carbon source, nitrogen source, inorganic salts and other elements to supply their growth and development,..."Please rephrase this as it is not specific to this species. Every living cell behaves the same way - it uses available nutrients to produce energy and synthesize the compounds it needs.
Response 2: Thank you for your valuable advice, the expert opinion has been adopted.(L34-36)
Comment 3:Please check the scientific names of the mentioned strains, Latin in italics.
Response 3: Thank you for your comments.
We have checked the scientific names of all strains in the full text and revised them,see the L47,53,386,387,391,392,415,436,444,461,481 and 482 and we apologize for our mistakes.
Comment 4:"...wheat is more sensitive to Mn...".Did you mean that wheat is more sensitive to the lack of Mn? Or on an inadequate amount of Mn?
Response 4: Thank you for your comments.
We have adjusted this section.(L65-67)
Comment 5:"Wangfound","V.volvaceacorrelation"Please separate the words.
Response 5: Thank you for your comments.
We have separated the words.(L67,340)
Comment 6:Please change "Mn, Cu "to "Mn and Cu".
Response 6: Thank you for your comments.We have changed it.(L69)
Comment 7:Please verify this claim? Is it a matter of substrate?
Response 7: Thank you for your comments.
We have verified this claim and fixed it.(L71)
Comment 8:"This study can provide theoretical reference and technical support for the rejuvenation of degenerated strains of edible mushrooms such as V.volvacea " .Is this sentence very similiar to the last one in the Abstract? Please, reconsider the last paragraph of the part Introduction.
Response 8: Thank you for your comments.We have made adjustments.(L76-77)
Comment 9:"T10: obtained by repeated tissue isolation for 10 times, with a signifificant decrease in yield; T19: obtained by repeated tissue isolation for 19 times, with a loss of fruiting body ability".Please explain in more detail how you obtained strains T10 and T19? Clarify obtaining the desiring traits. This is very important because the research is based on selected traits of the strains.
Response 9: Thank you for your valuable advice, the expert opinion has been adopted.(L84-90)
- volvaceasubcultured strains (T10 and T19) were obtained through the successive tissue isolation subculturing method. Briefly, T0 was cultivated, and an egg-shaped stage fruiting body was obtained. The fruiting body was cut, and a small piece from the junction of the stipe and the cap was used to generate the 1st generation strain, labeled as T1. T1 was cultivated, and the obtained fruiting body was used to obtain the 2nd generation strain, labeled as T2. Likewise, T1-T19 strains were obtained using 19 consecutive cultivations ( Figure 1). T10 and T19 were selected as experimental strains.
Figure 1. The tissue isolation process for succession strains is shown in corresponding images. (A) Original strains (T0), (B) Activation of strains, (C) Seed cultivation, (D) The original base, (E) Egg-shaped period, (F), Tissue separation of the fruiting body, (G) Tissue culture, (H) Strains preservation (T1-T19).
Comment 10:Please delete the repetition.
Response 10: Thank you for your comments.
We have deleted them.(L108,113,116,142,388,395 and 463)
Comment 11:Please change "petri"to"Petri", "protoplast " to "primordia ","primordial" to "primordia","the" to "The".
Response 11: Thank you for your comments.
We have revised these sentences.(L122,243,252 and 285)
Comment 12:"This study can provide theoretical reference and technical support for the rejuvenati on research of degenerated strains of V. volvacea and other edible fungi".Please redefine this statement, considering that the same sentence exists at the end of the abstract.
Response 12: Thank you for your comments.
We have revised these sentences.(L531-532)
Comment 13:"Na2SeO3",Please write the formula correctly.
Response 13: Thank you for your comments.
We have revised this word.(L386)
Comment 14: "GSH " ,"GSSG " and "PDB ",Please give the full names with the abbreviations in parentheses.
Response 14: Thank you for your comments.We have gived the full names with the abbreviations in parentheses.(L168-169,175)
Comment 15:"glucosidase activity(BGL), laccase (Lac), manganese peroxidase (MnP), hemicellulase ,(HMC) and xylanase (Xyl)".What? Is this sentence incomplete?
Response 15: Thank you for your comments.
We have made adjustments.(L161)
Comment 16:Please give an explanation below equation for ODck.
Response 16: Thank you for your valuable advice, the expert opinion has been adopted.(L137)
Thank you so much!
